computational chemistry/chemical physics/cybernetics

Belousov–Zhabotinsky reaction, oscillations, temperature-controlled, particle-coated droplets

**Author for correspondence:**
Andrew Adamatzky
e-mail: andrew.adamatzky@uwe.ac.uk

# Thermal switch of oscillation frequency in Belousov–Zhabotinsky liquid marbles

Andrew Adamatzky[1], Claire Fullarton[1], Neil Phillips[1], Ben De Lacy Costello[1,2] and Thomas C. Draper[1]

[1]Unconventional Computing Laboratory, Department of Computer Science and Creative Technologies, and [2]Institute of Biosensing Technology, Centre for Research in Biosciences, University of the West of England, Bristol BS16 1QY, UK

 AA, 0000-0003-1073-2662; CF, 0000-0003-0640-0497; TCD, 0000-0001-6466-8796

External control of oscillation dynamics in the Belousov–Zhabotinsky (BZ) reaction is important for many applications including encoding computing schemes. When considering the BZ reaction, there are limited studies dealing with thermal cycling, particularly cooling, for external control. Recently, liquid marbles (LMs) have been demonstrated as a means of confining the BZ reaction in a system containing a solid–liquid interface. BZ LMs were prepared by rolling $50 \, \mu l$ droplets in polyethylene (PE) powder. Oscillations of electrical potential differences within the marble were recorded by inserting a pair of electrodes through the LM powder coating into the BZ solution core. Electrical potential differences of up to $100 \, mV$ were observed with an average period of oscillation *ca* $44 \, s$. BZ LMs were subsequently frozen to $-1°C$ to observe changes in the frequency of electrical potential oscillations. The frequency of oscillations reduced upon freezing to $11 \, mHz$ cf. $23 \, mHz$ at ambient temperature. The oscillation frequency of the frozen BZ LM returned to $23 \, mHz$ upon warming to ambient temperature. Several cycles of frequency fluctuations were able to be achieved.

## 1. Introduction

Space–time dynamics of oxidation wavefronts, including target waves, spiral waves, localized wave-fragments and combinations of these, in a non-stirred Belousov–Zhabotinsky (BZ) medium [1,2] have been used to implement information processing since seminal papers by Kuhnert and co-workers [3,4]. The spectrum of unconventional computing devices prototyped with BZ reaction is rich. Examples include image processing and memory [5], diodes [6], geometrically constrained logical gates [7], controllers

for robots [8], wave-based counters [9], neuromorphic architectures [10–13] and binary arithmetical circuits [14–16].

While most of BZ computing devices use the presence of a wavefront in a selected locus of space as a manifestation of logical TRUE, there is a body of works on information coding with frequencies of oscillations. Thus, Gorecki et al. [17] proposed to encode TRUE as high frequency and FALSE as low frequency: OR gates, NOT gates and a diode have been realized in numerical models. Other results in BZ frequency-based information processing include frequency transformation with a passive barrier [18], frequency band filter [19] and memory [20]. Using frequencies is in line with current developments in oscillatory logic [21], fuzzy logic [11], oscillatory associated memory [22] and computing in arrays of coupled oscillators [23,24]. Therefore, frequencies of oscillations in BZ media will be the focus of this paper.

Most prototypes of BZ computers involve some kind of geometrical constraining of the reaction: a computation requires a compartmentalization. An efficient way to compartmentalize BZ medium is to encapsulate it in a lipid membrane [25,26]. This encapsulation enables the arrangement of elementary computing units into elaborate computing circuits and massive-parallel information processing arrays [27–30]. BZ vesicles have a lipid membrane and therefore have to reside in a solution phase, typically oil, and they are susceptible to disruption of the lipid vesicles through natural ageing and mechanical damage. Thus, potential application domains of the BZ vesicles are limited. This is why in the present paper we focus on liquid marbles (LMs), which offer us capability for 'dry manipulation' of the compartmentalized oscillatory medium. LMs also provide the possibility for active transport processes [31] which is not easily possible with vesicles, e.g. manipulating LMs with magnets [32,33], mechanically [34], electrostatically [35], pressure gradients [36], change in pH [37].

The LMs, proposed by Aussillous and Quéré in 2001 [38], are liquid droplets coated by hydrophobic particles at the liquid/air interface. The LMs do not wet surface and therefore can be manipulated by a variety of means [34], including rolling, mechanical lifting and dropping, sliding and floating [39–41]. The range of applications of LMs is huge and spans most fields of life sciences, chemistry, physics and engineering [31,42–45]. Recently, we demonstrated that the BZ reaction is compatible with typical LM chemistry: BZ–LMs support localized excitation waves, and non-trivial patterns of oscillations are evidenced in ensembles of the BZ LMs [46].

Oscillations in the BZ reaction media can be controlled by varying the concentrations of chemical species involved in the reaction, and with light [47,48], mechanical strain [49] and temperature [50–54]. While a number of high-impact results on the thermal sensitivity have been published, the topic still remains open and of utmost interest. Moreover, in LMs we might have difficulties in controlling the reaction with illumination because most types of hydrophobic coating are not perfectly transparent and absorb wavelengths of light important for exerting control over the BZ reaction. This is why in the present manuscript we focus on thermal control and tuning of the oscillations.

Temperature sensitivity of the BZ reaction was initially substantially analysed by Blandamer & Morris [50] who, in 1975, showed a dependence of the frequency of oscillations of a redox potential in a stirred BZ reaction with a change in temperature. Periods of oscillations reported were 190 s at 25°C, 70 s at 35°C, and 40 s at 45°C. In 1988, Vajda et al. [51] demonstrated that temporal oscillations of a BZ mixture persist in a frozen aqueous solution at −10°C to −15°C. By tracing $Mn^{2+}$ ion signal amplitude, they showed that the frozen BZ solutions oscillate three times, at −10°C, and 11 times, at −15°C, faster than liquid phase BZ. The oscillation frequency increase has been explained by the formation of crystals and interfacial phenomena during freezing. This might be partly supported by experiments with chlorite–thiosulphate system frozen to −34°C [55]. There, a velocity of wavefronts is increased because en route to total freezing the reaction occurs only in the thin liquid layer, at the periphery of the solid domain, where concentrations of chemicals are temporarily higher. In 2001, Masia et al. [52] monitored oscillations in non-stirred BZ in a batch reactor of $4 \, cm^3$ by the solution absorbency at 320 nm. The reactor was kept at various temperatures through thermostatic control. They reported periodic oscillation at temperatures 0°C–3°C, quasi-periodic at 4°C–6°C and chaotic at 7°C–8°C. Bánsági et al. [54] experimentally demonstrated that by increasing temperature from 40°C to 80°C it is possible to obtain oscillations of frequency over 10 Hz; they also showed that the frequency of oscillations grows proportionally to temperature (in the range studied). Ito et al. [53] reported linear dependence of an oscillation period—of polymers impregnated with BZ—from temperature in the range 5°C–25°C.

We establish an electrical interface with BZ LMs by piercing them with a pair of electrodes. This is done for two reasons. First, the coating of LMs is usually non-transparent, therefore conventional optical means of recording oxidation dynamics would not be sufficient. In addition, marbles are three-dimensional structures and there is evidence that they support complex three-dimensional waves,

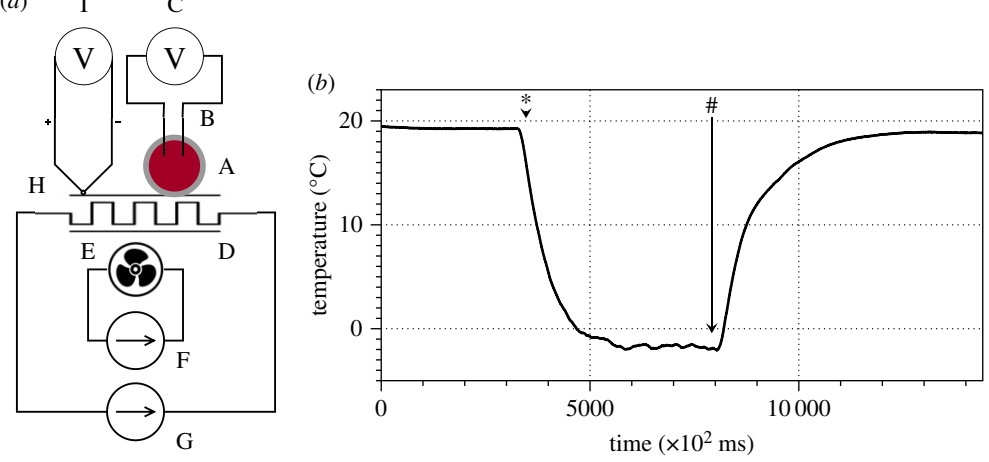

**Figure 1.** Experimental set-up. (*a*) A scheme of the set-up: A, BZ LM; B, a pair of electrodes; C, Pico ADC-24 logger; D, Peltier element; E, fans; F, power supply for fans; G, power supply for the Peltier element; H, thermocouple; I, TC-08 thermocouple data logger. (*b*) Dynamics of temperature on the surface of the Petri dish when the Peltier element is powered by 7 V. The moment of power on is shown by '*' and off by '#'.

therefore, electrodes positioned within the marble potentially allow the three-dimensional oscillation dynamics to be mapped, whereas imaging is difficult to interpret from a three-dimensional standpoint. Second, our ultimate goal is to implement an unconventional computing device with BZ LMs. Such devices rarely stand alone but are usually interfaced with conventional electronics, thus electrical recording seemed to be most appropriate.

## 2. Methods

BZ LMs were produced by coating droplets of BZ solution with ultra high-density polyethylene (PE) powder (Sigma Aldrich, CAS 9002-88-4, Product Code 1002018483, particle size 150 µm). The BZ solution was prepared using the method reported by Field & Winfree [56], omitting the surfactant Triton X. The 18 M Sulfuric acid $H_2SO_4$ (Fischer Scientific), sodium bromate $NaBrO_3$, malonic acid $CH_2(COOH)_2$, sodium bromide NaBr and 0.025 M ferroin indicator (Sigma Aldrich) were used as received. Sulfuric acid (2 ml) was added to deionized water (67 ml), to produce 0.5 M $H_2SO_4$; $NaBrO_3$ (5 g) was added to the acid to yield 70 ml of stock solution (0.48 M).

Stock solutions of 1 M malonic acid and 1 M NaBr were prepared by dissolving 1 g in 10 ml of deionized water. In a 50 ml beaker, 0.5 ml of 1 M malonic acid was added to 3 ml of the acidic $NaBrO_3$ solution; 0.25 ml of 1 M NaBr was then added to the beaker, which produced bromine. The solution was set aside until it was clear and colourless (*ca* 3 min) before adding 0.5 ml of 0.025 M ferroin indicator.

BZ LMs were prepared by pipetting a 75 µl droplet of BZ solution, from a height of *ca* 2 mm onto a powder bed of PE, using a method reported previously [46]. The BZ droplet was rolled on the powder bed for *ca*. 10 s until it was fully coated with powder.

A scheme of experimental set-up is shown in figure 1*a*. An LM was placed in a Petri dish (35 mm diameter) and pierced with two iridium-coated stainless steel electrodes (sub-dermal needle electrodes with twisted cables (SPES MEDICA SRL Via Buccari 21 16153 Genova, Italy). Electrical potential difference between electrodes was recorded with a Pico ADC-24 high-resolution data logger (Pico Technology, St Neots, Cambridgeshire, UK), sampling rate 25 ms.

A Petri dish with LM was mounted to a Peltier element (100 W, 8.5 A, 20 V, 40 × 40 mm, RS Components Ltd, UK ), which in turn was fixed to an aluminium heat sink, with Silver CPU Thermal Compound, cooled by two 12 V fans (powered separately from the Peltier element). Temperature at the Peltier element was controlled via RS PRO Bench Power Supply Digital (RS Components Ltd, Corby, Northants, UK). Temperature at the bottom of the Petri dish was monitored using TC-08 thermocouple data logger (Pico Technology, St Neots, Cambridgeshire, UK), sampling rate 100 ms. A typical cooling rate was −1°C per 10 s, and warming rate +1°C per 20 s, exact shape of the functions is shown in figure 1*b*.

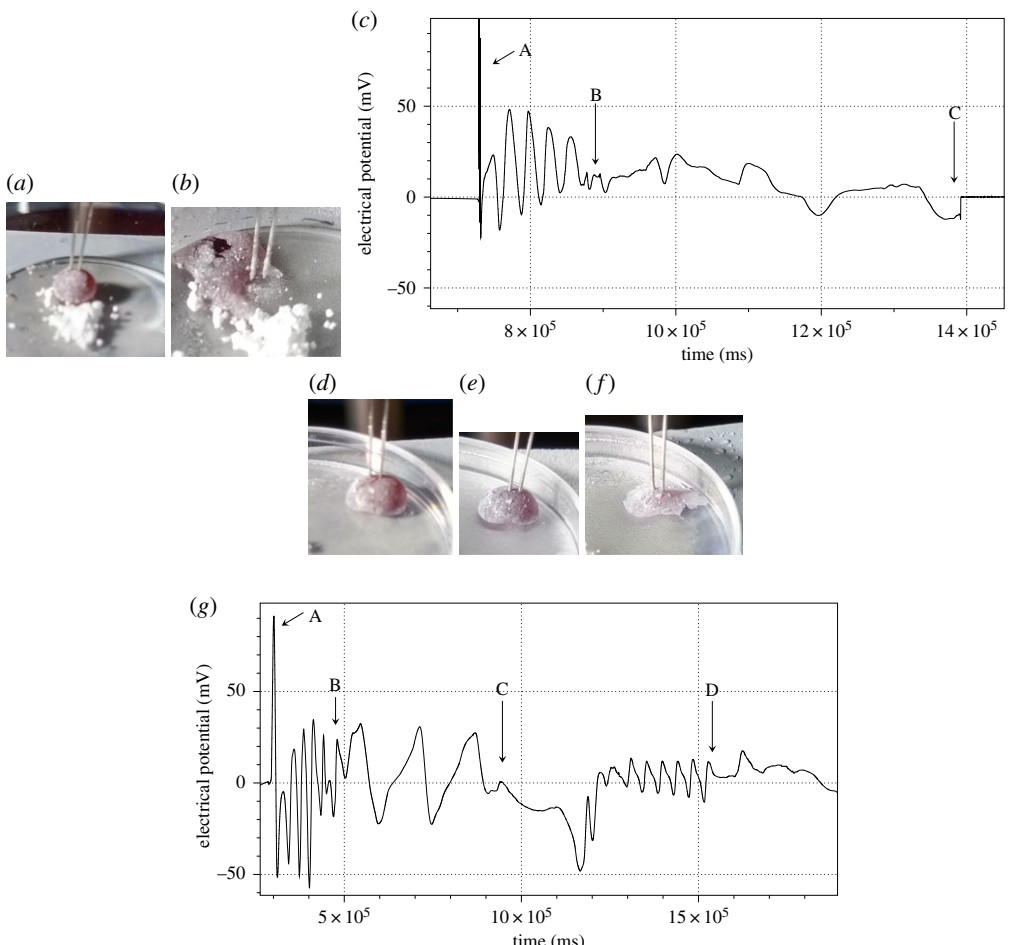

**Figure 2.** ($a-c$) LM bursts at first freezing. ($d-g$) LM burst at second freezing. ($a$) Marble at the beginning of experiment. ($b$) Marble burst at some point of freezing. ($c$) Plot of oscillations: A, the marble is stimulated with a silver wire; B, oscillations started, Peltier element is switched on; C, marbles cools downs, eventually the marbles bursts. ($d$) LM at the beginning of experiment. ($e$) Cooled-down LM. ($f$) LM bursts and spreads at the second round of freezing. ($g$) Dynamics of electrical potential: A, marble is stimulated by a silver wire for 2–3 s; B, Peltier element is switched on; C, Peltier is switched off; D, Peltier is switched on again.

## 3. Results

Temperatures on the surface of the Petri dish below $-2°$C usually result in a burst LM. Two examples are illustrated in figure 2. An intact LM (figure 2$a$) shows oscillations with average period 26 s (between 'A' and 'B' in figure 2$c$). When cooling is started ('B' in figure 2$c$) oscillations quickly become low-frequency low-amplitude irregular, average period 49 s. Eventually, the LM bursts ('C' in figure 2$c$) and its cargo is relocated away from the electrodes (figure 2$b$). In the scenario shown in figure 2$d-g$, LM undergoes two instances of freezing. First time, marked 'B' in figure 2$g$ the LM (figure 2$d$) survives being cooled down with just slight change in shape (figure 2$e$). Period of oscillations increases from 28 s in intact LM to 162 s in cooled-down LM (period between 'B' and 'C' in figure 2$g$). After Peltier is switched off (moment 'C' in figure 2$g$), the LM resumes high-frequency oscillations, frequency 42 s, but with lower amplitude. The LM does not survive second round of freezing ('D' in figure 2$g$) and bursts, while still wetting the electrodes (figure 2$f$). More examples of electrical potential dynamics for temperatures causing LM bursting are shown in figure 3. The temperature of $-2°$C is critical, in that over 70% of LMs burst and did not survive second round of freezing. Therefore, in further experiments the LMs were cooled down to $-1°$C.

Patterns of oscillations of LM cooled down to $-1°$C show a high degree of polymorphism (figure 4) in amplitudes. Changes in frequencies are in table 1. If we ignore the first example (figure 4$a$), then we have average $p = 44.4$ ($\sigma(p) = 12.5$), average $p^* = 92$ ($\sigma(p^*) = 28.6$) average $p^*/p = 2.1$ ($\sigma(p^*/p) = 0.5$).

In the experiments shown in figure 4$a-e$, LMs were kept cooled until the end of the experiments. In the experiment shown in figure 4$f$, cooling was started after 1310 s of the experiment, the Peltier was

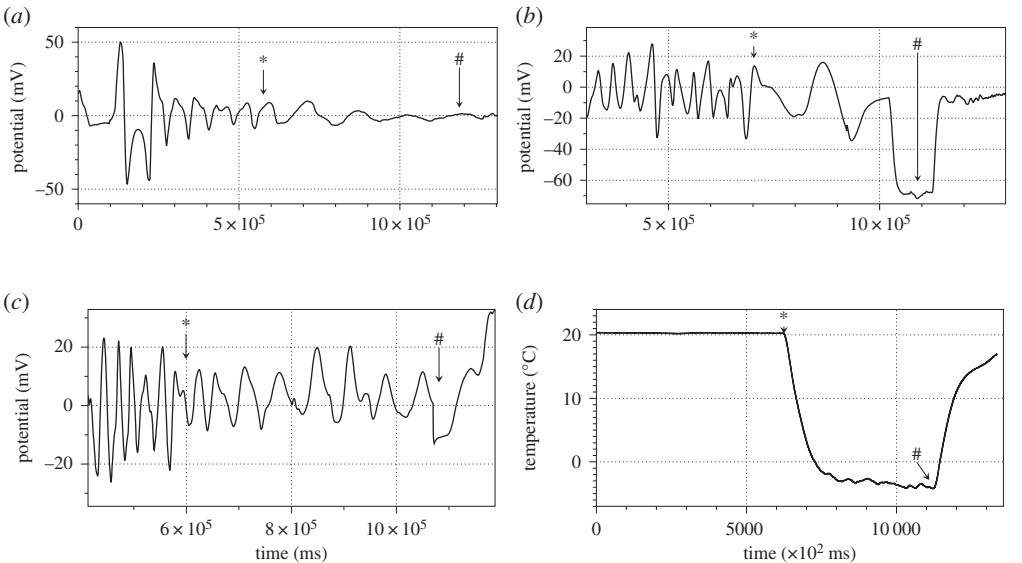

**Figure 3.** Dynamics of electrical potential of LM cooled, temperature at the bottom of the Petri dish, down to (a) $-4°$C, (c) $-3°$C, (d) $-2°$C. Moment when Peltier element is switched on is shown by '*' and off by '#'. (d) Temperature log corresponding to experiments (a).

**Figure 4.** Dynamics of electrical potential of BZ LM subjected to cooling down to $-1°$C and warming up. Moments when the Peltier element was switched on are shown by '*' and off by '#'. Moments when electrodes are inserted in the LM are shown by '!'.

**Table 1.** Effect of cooling to $-1°C$ on a period of electrical potential oscillations of BZ LM: $p$ is a period of electrical potential oscillation of an LM at ambient temperature, $p^*$ is a period of electrical potential oscillations of the cooled LM.

| plot | $p$, s | $p^*$, s | $\frac{p^*}{p}$ |
| --- | --- | --- | --- |
| figure 4a | 61 | 336 | 5.5 |
| figure 4b | 59 | 126 | 2.1 |
| figure 4c | 56 | 138 | 2.5 |
| figure 4d | 22 | 67 | 3 |
| figure 4e | 39 | 86 | 2.2 |
| figure 4f | 47 | 74 | 1.6 |
| figure 4g | 29 | 67 | 2.3 |
| figure 4h | 39 | 86 | 2.2 |

switched off after 2254 s, and cooling was repeated at 2690 s. Intact LM oscillated with average period 47 s at first phase of the experiment. Cooled LM oscillated with period 74 s. The period became 29 s after the warming. Second cooling increased the period to 67 s. Thus, we have an increase of 1.6 times during first cooling cycle and by 2.3 times during the second cooling cycle. In the experiment illustrated in figure 4g, oscillations were arrested by cooling yet restarted when the LM was warmed. Period of oscillations before cooling was 47 s, and after oscillations restarted after cooling was 39 s. Repeated cooling did not arrest oscillations yet increased the oscillation period 2.2 times to 86 s. In the experiment shown in figure 4h, we cooled an LM for short periods of time (199 and 288 s) and did not observe any substantial changes in periods of oscillation, after the first freezing cycle. The average periods were changing as follows 46 s → 92 s → 98 s → 98 s → 98 s.

To summarize, average period of oscillations of a BZ LM doubles from 44 s to 92 s when the LM is cooled down to $-1°C$. The frequency of oscillations is restored after cooling is stopped. The amplitude of oscillations may increase or decrease as a result of cooling. Sometimes the oscillations can be completely arrested yet resume after warming.

# 4. Discussion

Why are oscillations of electrical potential observed? The oxidation of malonic acid by bromate ions in acidified solution is catalysed by ferroin ions. Ferroin ions $[\mathrm{Fe}(\mathrm{ox}-phen)_3]^{2+}$ are oxidized to their ferric derivatives $[\mathrm{Fe}(\mathrm{o}-phen)_3]^{3+}$. The ratios of ferroin to ferric ions and bromide ions oscillate in time. This is reflected in the oscillations of the electrical potential recorded from the LM. If the BZ solution in an LM was mixed, then global oscillations would occur, resulting in the potential at both electrodes being the same and therefore no electrical oscillations could be observed. However, the solution is not mixed, therefore waves of oxidation emerge spontaneously, or are induced when the LM is pierced by electrodes, or induced by piercing with a silver wire (the silver catalyses a local reduction in bromate concentration, initiating the reaction). Therefore, the ratio of ferroin to ferric ions (and bromide ions) are changing only at the wavefront. Thus, when the wavefront passes the electrodes the electrical potential difference is observed.

Why are patterns of oscillations not always regular? This is because several oxidation waves, and even several generators/sources of oxidation waves, can coexist in a single LM. These waves can superimpose with each other, collide and annihilate in the result of the collisions, or produce localized wave-fragments. This rich dynamic of wavefronts is reflected in, sometimes, irregular patterns of oscillation. Let us illustrate further discussions with two-variable Oregonator equations [57,58],

$$\frac{\partial u}{\partial t} = \frac{1}{\epsilon}\left(u - u^2 - (fv + \phi)\frac{u - q}{u + q}\right) + D_u \nabla^2 u$$

$$\text{and} \quad \frac{\partial v}{\partial t} = u - v. \tag{4.1}$$

The variables $u$ and $v$ represent local concentrations of an activator, or an excitatory component of BZ system, and an inhibitor, or a refractory component. Parameter $\epsilon$ sets up a ratio of the time scale of

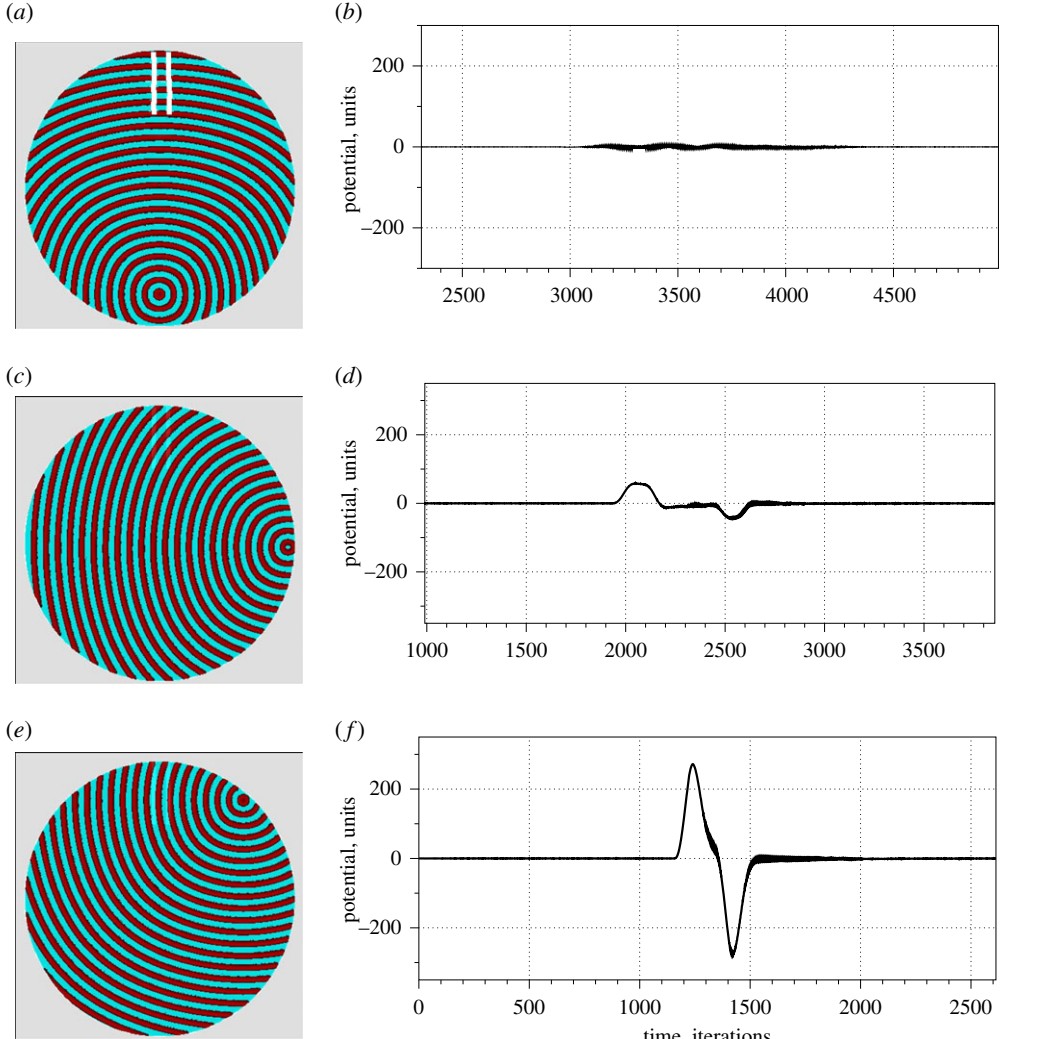

**Figure 5.** Time-lapse snapshots (*a,c,e*) and corresponding potential recorded at the electrodes (*b,d,f*) of a single wave initiated at southern edge of the droplet (*a,b*), eastern edge of the droplet (*c,d*), northeastern edge of the droplet (*e,f*). The time-lapse snapshots were recorded at every 150th time step. We display sites with $u > 0.04$. Domains corresponding to the electrodes are shown by white rectangles in (*a*).

variables $u$ and $v$, $q$ is a scaling parameter depending on rates of activation/propagation and inhibition, and $f$ is a stoichiometric coefficient. We integrated the system using Euler method with five-node Laplace operator, time step $\Delta t = 0.001$ and grid point spacing $\Delta x = 0.25$, $\epsilon = 0.02$, $f = 1.4$, $q = 0.002$. We varied value of $\phi$ from the interval $\Phi = [0.05, 0.08]$, where constant $\phi$ is a rate of inhibitor production. $\phi$ represents the rate of inhibitor; this rate can be dependent on light, temperature or the presence of other chemical species. The parameter $\phi$ characterizes excitability of the simulated medium, i.e. the larger $\phi$ the less excitable the medium is. We represent BZ LM as a disc with a radius of 185 nodes. We represent electrodes as rectangular domains of the discs (see figure 5*a* and figure 7*a*) $\mathcal{E}_1$ and $\mathcal{E}_2$. We calculate the potential difference at each iteration $t$ as $\sum_{x \in \mathcal{E}_2} u_x^t - \sum_{x \in \mathcal{E}_1} u_x^t$.

Orientation of the wavefront passing the electrodes determines exact shape of the impulse recorded (figure 5). Assume a droplet is excitable everywhere. If a wavefront is perpendicular to the electrodes, e.g. a wave is generated at the southern edge of the droplet (figure 5*a*), the potential difference between electrodes at any moment of time will be near zero, a part of some noise (figure 5*b*). A wave originated at the eastern edge of a droplet enters electrodes at an obtuse angle (figure 5*c*). This is reflected in two spikes—one is positive potential and another is negative potential (figure 5*d*), there is a substantial distance between the spikes. If the wavefront propagates nearly parallel to the electrodes, e.g. when a wave is generated at northeast edge of the droplet (figure 5*e*), the action-like potential is recorded (figure 5*f*), which shape imitates distinctive depolarization, repolarization and hyperpolarization phases of a biological action potential. In experiments, we always observed oscillation. The shape of the impulses

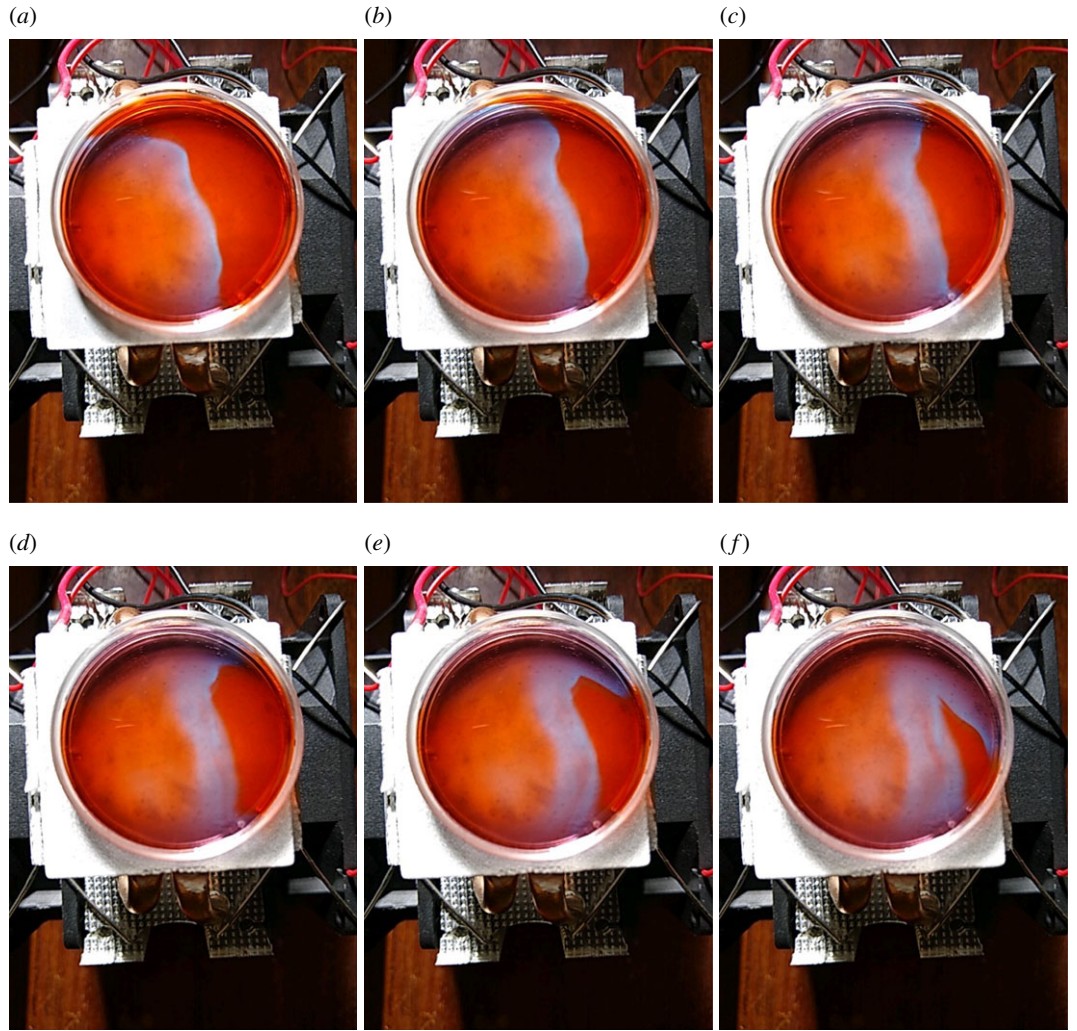

(a)    (b)    (c)

(d)    (e)    (f)

**Figure 6.** Time lapse photos of the propagation of an oxidation wavefront in a thin layer of BZ medium on the freezing Peltier element. Time from the start of recording is (a) 90 s, (b) 126 s, (c) 150 s, (d) 174 s, (e) 198 s and (f) 246 s.

was nearly the same—subject to deviations—in all experiments. This implies that the wavefront travels not in the volume of BZ LM but along the surface of the LM. Thus, the wavefront passes electrodes being nearly parallel to them.

Why does the frequency of oscillations decrease on cooling? Temperature changes the rate of the reaction which consumes the inhibitor of the auto-catalytic $Br^-$ [50] species. When the temperature decreases, the rate of consumption of $Br^-$ also decreases, which increases the time necessary for the reaction to enter its auto-catalytic step. The enlargement of the refractory tail reduces the number of wavefronts that can be fitted in a limited space. Thus less waves pass electrodes in a given period of time. This is reflected in a reduced frequency of oscillations. The mechanism is illustrated in experiments with a thin-layer BZ medium shown in figure 6 and simulation with Oregonator model in figure 7. A 35 mm Petri dish was placed on the freezing set-up (figure 1), and the element was chilled to $-7°C$. The BZ medium did not freeze but its temperature dropped to near $0°C$. The cooling was reflected in the enlarged tail of the excitation wavefront, it doubled in width from 2.5 mm (figure 6a) to 4.7 mm (figure 6e) in just over 3 min. In modelling the BZ medium (figure 7), we position electrodes in the north of the droplet and assume a self-excitation loci near the edge at the east of the droplet (figure 7a) and that waves propagate only near the surface (i.e. only part of 370-nodes-wide disc with $r > 150$ is excitable). The excitable loci $\mathcal{L}$ have values $u_x = 1$, $x \in \mathcal{L}$, at every iteration of the numerical integration; however, waves are generated only with some intervals. Distance between wavefronts increases with decrease of excitability, increase of $\phi$ from 0.01 (figure 7b) to 0.07 (figure 7e). This is reflected in decreasing of oscillation frequency of the potential difference recorded at the electrodes (figure 7f–i). The shapes of impulses in figure 7i strikingly resemble shapes of

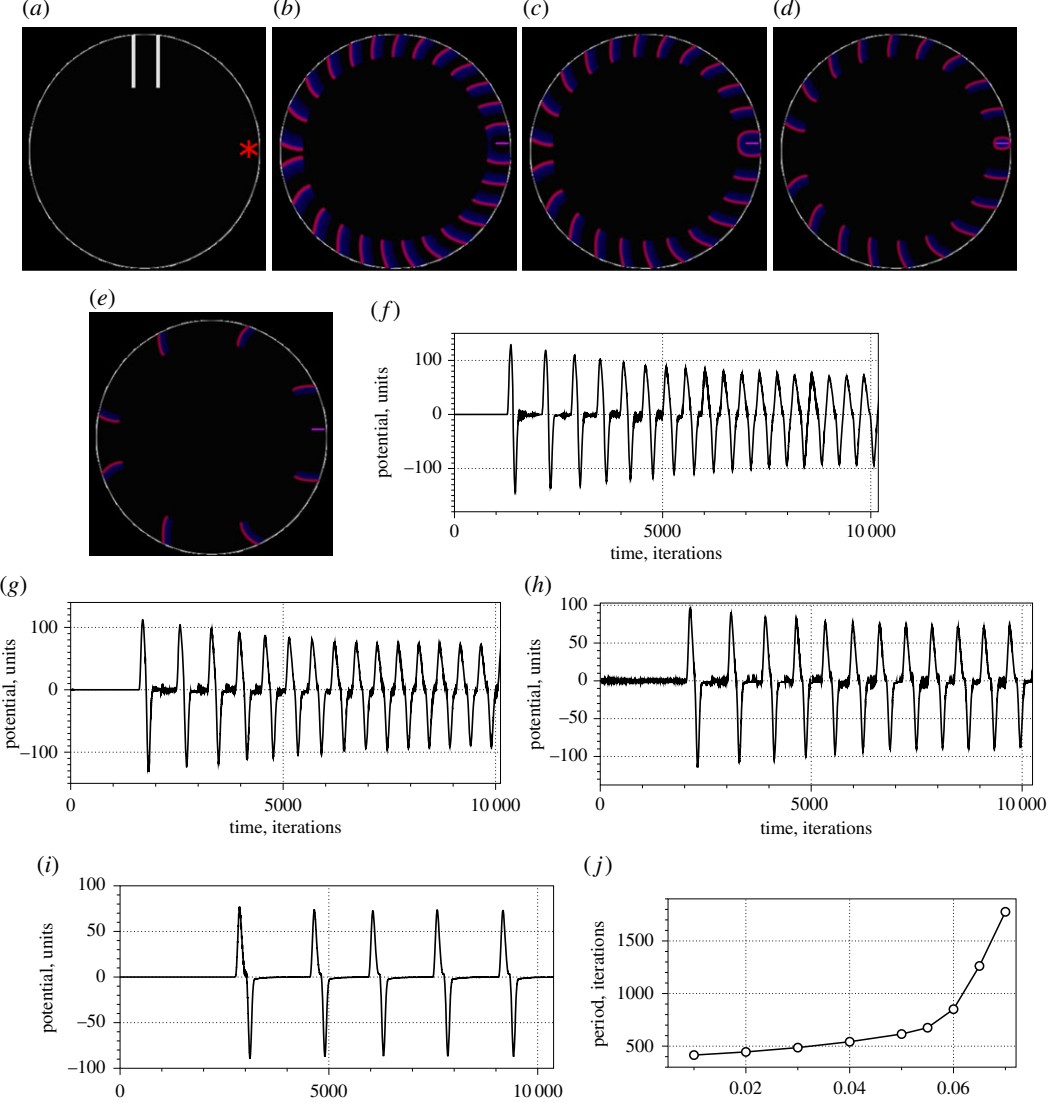

**Figure 7.** Modelling wavefront propagation and frequency of wave generation for various values of excitability $\phi$. (*a*) Position of electrodes, auto-excitation loci $\mathcal{L}$ is shown by star. (*b–e*) Snapshots of the medium showing trains of the wavefronts. (*f–i*) Potential difference recorded on the electrodes. (*j*) Period of the potential oscillations versus excitability $\phi$. (*a*) $\phi = 0.01$, $t = 10^4$, (*b*) $\phi = 0.01$, $t = 10^4$, (*c*) $\phi = 0.03$, $t = 10^4$, (*d*) $\phi = 0.05$, $t = 10^4$, (*e*) $\phi = 0.07$, $t = 3 \times 10^4$, (*f*) $\phi = 0.01$, (*g*) $\phi = 0.03$, (*h*) $\phi = 0.05$, (*i*) $\phi = 0.07$.

experimentally recorded impulses in figure 4*a*. The dependence of oscillation period on excitability $\phi$ is linear for $\phi \in [0.01, 0.05]$ and cubic for $\phi \in [0.05, 0.07]$ (figure 7*j*).

How long can the oscillations last? In our experiments, the oscillations in a 50 µl LM lasted up to an hour. The amplitude decreases with time due to exhaustion of catalyst in the droplet; however, the most typical cause of oscillations ceasing was breakage of the LMs. Generally, repeated cycles of freezing and warming caused disruption of the hydrophobic particle 'skin' of an LM, resulting in the cargo being spilled.

How can the observed phenomena be used in unconventional computing? As Horowitz and Hill mention in their famous 'The Art of Electronics'—'A device without an oscillator either does not do anything or expects to be driven by something else (which probably contains an oscillator)' [59]. We produced a chemical analogue of an electronic temperature-sensitive oscillator: an oscillator circuit for sensing and indicating temperature by changing oscillator frequency with temperature [60,61]. Future BZ computing devices will be hybrid chemical-electronic devices, needing components to generate waveforms. The BZ LMs *per se* are sources of (relatively) regular space pulses. We experimentally demonstrated that the frequency of the pulses can be switched from high to low by freezing the BZ

LMs. This realization could be used in future large-scale ensembles of BZ LMs which approximate fuzzy-logic many-argument functions, where inputs are represented by temperature gradients, and outputs are dominating frequencies of the oscillations in the ensembles. To control frequencies in an ensemble of BZ LMs [46], we can use small Peltier elements, the size of which is enough just to fit a single marble, e.g. Peltier Module, 1.3 W, 2.2 A, 0.9 V, $6 \times 6$ mm (each of the modules can be automatically controlled via Arduino device). Additional future challenges would include a comparison between a surface simulation and the surface reconstruction, implementation of experiments on freezing microdroplets, as inspired by Wang *et al*. [62], freezing of photo-sensitive BZ LMs in combination with intermittent illumination.

Data accessibility. The data supporting the findings of this study are available at http://doi.org/10.5281/zenodo.2533350.
Authors' contributions. A.A., C.F., T.C.D., N.P. and B.D.L.C. undertook the research and wrote the manuscript.
Competing interests. We declare we have no competing interests.
Funding. This research was supported by the EPSRC with grant EP/P016677/1 'Computing with Liquid Marbles'.
Acknowledgements. A.A. thanks Jitka Čejková (University of Chemistry and Technology Prague, Czech Republic) for introducing him to liquid marbles in 2016.

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
