## [Reviewer comments · Royal Society Open Science]

Review History

RSOS-190078.R0 (Original submission)

Review form: Reviewer 1 (Kyle Harrington)

Is the manuscript scientifically sound in its present form?

Yes

Are the interpretations and conclusions justified by the results?

Yes

Is the language acceptable?

Yes

Is it clear how to access all supporting data?

No

Do you have any ethical concerns with this paper?

No

Have you any concerns about statistical analyses in this paper?

I do not feel qualified to assess the statistics

Recommendation?

Accept with minor revision (please list in comments)

Comments to the Author(s)

This is an important paper because it reveals a new tractable approach to controlling the behavior of BZ. Furthermore, by focusing their efforts on the BZ liquid marble formulation, the authors have developed a pathway for control that can be controlled in a modular fashion.

Do the authors have any consideration about what would happen if microdroplets were frozen (e.g., Wang et al 2016 EPJ)? The ratio of change in length scale would be notably larger and could be sufficient to toggle information transmission between inhibitory coupling and activating coupling.

Have the authors attempted to combine this method with photosensitive BZ? Can other input methods be used at the same time as freezing?

Oscillations look more irregular after freezing. How does this impact the potential to use freezing/frozen liquid marbles in a computational circuit?

How does freezing affect the longevity of oscillatory behavior in LMs? That is, do frozen droplets oscillate the same number of times as non-frozen LMs?

It would be interesting to see some numbers that describe the variability of LM behavior after freezing. How consistent and/or predictable is their behavior?

It would also be interesting to see a comparison between the surface simulations and an surface reconstruction. However, this seems likely to be more appropriate for a subsequent effort.

Simulation code is used, but no link is provided. The authors have shared their code in previous papers, so this is clearly a simple oversight.

Review form: Reviewer 2**Is the manuscript scientifically sound in its present form?**

Yes

Are the interpretations and conclusions justified by the results?

Yes

Is the language acceptable?

Yes

Is it clear how to access all supporting data?

Yes

Do you have any ethical concerns with this paper?

No

Have you any concerns about statistical analyses in this paper?

No

Recommendation?

Accept with minor revision (please list in comments)

Comments to the Author(s)

The manuscript entitled "Thermal switch of oscillation frequency in Belousov-Zhabotinsky liquid marbles" authored by A. Adamatzky, C. Fullarton, T. Draper, N. Philips and B. de Lacy Costello describes temperature effect on period of oscillations in Belousov-Zhabotinsky (BZ) droplets coated with polyethylene powder (named here as liquid marbles (LMs)). Authors claim that this phenomena can be useful as a control method in information processing devices based on BZ LMs. As attention towards new unconventional computing methods is growing the topic seems interesting. The manuscript needs revision before publication. It might be suitable for publication in Royal Society Open Science upon minor changes

1. In the first section authors mention that "BZ-LMs also provide the possibility for active transport processes which is not easily possible with vesicles". It would be helpful if the authors explain what they mean by active processes in BZ-LMs (and why it is hard to realise with vesicles) or give at least some literature reference.
2. For implementing some useful computing functionality larger number of the BZ LM's is required. With the presented thermal control method I see no possibility of oscillation control of individual LM's in an ensemble. Another question is how temperature difference influence affects mechanical stability of BZ LMs? Do authors expect other problems if the present method is applied for a structure similar to this from ref. 41? I think it would be fair to the readers to discuss these and other potential problems with thermal control of BZ LMs.
3. As described in caption of Fig. 2c in this experiment oscillations were induced with a silver wire. Please add information about purpose of silver wire stimulation to Methods section.
4. In Fig. 3d it would be helpful if the moments of time at which Peltier was on and off are also marked (as in Fig 3a with * and # symbols).
5. All the figures would be easier to follow if the time axis are changed to seconds, especially that all of the times and period in the text are given in seconds.
6. There are some small issues, which I've noticed, and which could help the Authors with polishing the manuscript:
 - phrase repetitions
 - line 36 page 5 "are as follows are analysed".
 - Line 43 page 7 "This is is reflected"
 - According to caption of Fig.4a it was kept cool until the end of experiment. Why # symbol is visible at $t = 900$ s?

Decision letter (RSOS-190078.R0)

11-Mar-2019

Dear Dr Adamatzky

On behalf of the Editors, I am pleased to inform you that your Manuscript RSOS-190078 entitled "Thermal switch of oscillation frequency in Belousov-Zhabotinsky liquid marbles" has been accepted for publication in Royal Society Open Science subject to minor revision in accordance with the referee suggestions. Please find the referees' comments at the end of this email.

The reviewers and handling editors have recommended publication, but also suggest some minor revisions to your manuscript. Therefore, I invite you to respond to the comments and revise your manuscript.

- Ethics statement

- Data accessibility

<http://datadryad.org/submit?journalID=RSOS&manu=RSOS-190078>

- Competing interests

- Authors' contributions

AB carried out the molecular lab work, participated in data analysis, carried out sequence alignments, participated in the design of the study and drafted the manuscript; CD carried out

the statistical analyses; EF collected field data; GH conceived of the study, designed the study, coordinated the study and helped draft the manuscript. All authors gave final approval for publication.

- Acknowledgements

- Funding statement

Because the schedule for publication is very tight, it is a condition of publication that you submit the revised version of your manuscript before 20-Mar-2019. Please note that the revision deadline will expire at 00.00am on this date. If you do not think you will be able to meet this date please let me know immediately.

- 1) A text file of the manuscript (tex, txt, rtf, docx or doc), references, tables (including captions) and figure captions. Do not upload a PDF as your "Main Document";
- 2) A separate electronic file of each figure (EPS or print-quality PDF preferred (either format should be produced directly from original creation package), or original software format);
- 3) Included a 100 word media summary of your paper when requested at submission. Please ensure you have entered correct contact details (email, institution and telephone) in your user account;
- 4) Included the raw data to support the claims made in your paper. You can either include your data as electronic supplementary material or upload to a repository and include the relevant doi within your manuscript. Make sure it is clear in your data accessibility statement how the data can be accessed;

5) All supplementary materials accompanying an accepted article will be treated as in their final form. Note that the Royal Society will neither edit nor typeset supplementary material and it will be hosted as provided. Please ensure that the supplementary material includes the paper details where possible (authors, article title, journal name).

on behalf of Dr Matjaz Perc (Associate Editor) and Miles Padgett (Subject Editor)
openscience@royalsociety.org

Reviewer comments to Author:

Reviewer: 1

Comments to the Author(s)

This is an important paper because it reveals a new tractable approach to controlling the behavior of BZ. Furthermore, by focusing their efforts on the BZ liquid marble formulation, the authors have developed a pathway for control that can be controlled in a modular fashion.

Do the authors have any consideration about what would happen if microdroplets were frozen (e.g., Wang et al 2016 EPJ)? The ratio of change in length scale would be notably larger and could be sufficient to toggle information transmission between inhibitory coupling and activating coupling.

Have the authors attempted to combine this method with photosensitive BZ? Can other input methods be used at the same time as freezing?

Oscillations look more irregular after freezing. How does this impact the potential to use freezing/frozen liquid marbles in a computational circuit?

How does freezing affect the longevity of oscillatory behavior in LMs? That is, do frozen droplets oscillate the same number of times as non-frozen LMs?

It would be interesting to see some numbers that describe the variability of LM behavior after freezing. How consistent and/or predictable is their behavior?

It would also be interesting to see a comparison between the surface simulations and an surface reconstruction. However, this seems likely to be more appropriate for a subsequent effort.

Simulation code is used, but no link is provided. The authors have shared their code in previous papers, so this is clearly a simple oversight.

Reviewer: 2

Comments to the Author(s)

The manuscript entitled "Thermal switch of oscillation frequency in Belousov-Zhabotinsky liquid marbles" authored by A. Adamatzky, C. Fullarton, T. Draper, N. Philips and B. de Lacy Costello describes temperature effect on period of oscillations in Belousov-Zhabotinsky (BZ) droplets coated with polyethylene powder (named here as liquid marbles (LMs)). Authors claim that this phenomena can be useful as a control method in information processing devices based on BZ LMs. As attention towards new unconventional computing methods is growing the topic seems interesting. The manuscript needs revision before publication. It might be suitable for publication in Royal Society Open Science upon minor changes

1. In the first section authors mention that "BZ-LMs also provide the possibility for active transport processes which is not easily possible with vesicles". It would be helpful if the authors explain what they mean by active processes in BZ-LMs (and why it is hard to realise with vesicles) or give at least some literature reference.
2. For implementing some useful computing functionality larger number of the BZ LM's is required. With the presented thermal control method I see no possibility of oscillation control of individual LM's in an ensemble. Another question is how temperature difference influence affects mechanical stability of BZ LMs? Do authors expect other problems if the present method is applied for a structure similar to this from ref. 41? I think it would be fair to the readers to discuss these and other potential problems with thermal control of BZ LMs.
3. As described in caption of Fig. 2c in this experiment oscillations were induced with a silver wire. Please add information about purpose of silver wire stimulation to Methods section.
4. In Fig. 3d it would be helpful if the moments of time at which Peltier was on and off are also marked (as in Fig 3a with * and # symbols).
5. All the figures would be easier to follow if the time axis are changed to seconds, especially that all of the times and period in the text are given in seconds.
6. There are some small issues, which I've noticed, and which could help the Authors with polishing the manuscript:
 - phrase repetitions
 - line 36 page 5 "are as follows are analysed".
 - Line 43 page 7 "This is is reflected"

- According to caption of Fig.4a it was kept cool until the end of experiment. Why # symbol is visible at $t = 900$ s?

Author's Response to Decision Letter for (RSOS-190078.R0)

See Appendix A.

Decision letter (RSOS-190078.R1)

27-Mar-2019

Dear Dr Adamatzky,

I am pleased to inform you that your manuscript entitled "Thermal switch of oscillation frequency in Belousov-Zhabotinsky liquid marbles" is now accepted for publication in Royal Society Open Science.

on behalf of Dr Matjaz Perc (Associate Editor) and Professor Miles Padgett (Subject Editor)
openscience@royalsociety.org

Appendix A

Dear Editors and Reviewers,

We thank you for your constructive comments, please find updates marked red in the PDF of the paper and details listed below.

Reviewer: 1

1. Do the authors have any consideration about what would happen if microdroplets were frozen (e.g., Wang et al 2016 EPJ)? The ratio of change in length scale would be notably larger and could be sufficient to toggle information transmission between inhibitory coupling and activating coupling.

Authors: We have added this as a future challenge in the Discussion as follows

Additional future challenges would include a comparison between a surface simulations and the surface reconstruction, implementation of experiments on freezing microdroplets, as inspired by \cite{wang2016configurable}.

2. Have the authors attempted to combine this method with photosensitive BZ? Can other input methods be used at the same time as freezing?

Authors: We did not attempt in reported experiments, this could be a topic of future studies.

3. Oscillations look more irregular after freezing. How does this impact the potential to use freezing/frozen liquid marbles in a computational circuit?

Authors: Irregularity of oscillations might add a noisy (or innovation component) well thought in the application on evolutionary optimisation and learning.

4. How does freezing affect the longevity of oscillatory behavior in LMs? That is, do frozen droplets oscillate the same number of times as non-frozen LMs?

Authors: We have no evidence that freezing increases longevity of BZ LMs.

5. It would be interesting to see some numbers that describe the variability of LM behavior after freezing. How consistent and/or predictable is their behavior?

Authors: Yes, indeed, we have in the paper values of standard deviation:

...then we have average $p=44.4$ ($\sigma(p)=12.5$), average $p^*=92$ ($\sigma(p^*)=28.6$) average $\frac{p^*}{p}=2.1$ ($\sigma(\frac{p^*}{p})=0.5$)

6. It would also be interesting to see a comparison between the surface simulations and an surface reconstruction. However, this seems likely to be more appropriate for a subsequent effort.

Authors: We have added this as a future challenge in the Discussion as follows:

Additional future challenges would include a comparison between a surface simulations and the surface reconstruction.

Reviewer: 2

1. In the first section authors mention that “BZ-LMs also provide the possibility for active transport processes which is not easily possible with vesicles”. It would be helpful if the authors explain what they mean by active processes in BZ-LMs (and why it is hard to realise with vesicles) or give at least some literature reference.

Authors: We have updated the text as follows:

LMs also provide the possibility for active transport processes~\cite{bormashenko2011liquid} which is not easily possible with vesicles, e.g. manipulating LMs with magnets~\cite{zhao2010magnetic,zhang2012remotely}, mechanically~\cite{ooi2015manipulation}, electrostatically~\cite{bormashenko2011electrically}, pressure gradients~\cite{bormashenko2010micropump}, change in pH~\cite{fujii2010ph}.

2. For implementing some useful computing functionality larger number of the BZ LM's is required. With the presented thermal control method I see no possibility of oscillation control of individual LM's in an ensemble. Another question is how temperature difference influence affects mechanical stability of BZ LMs? Do authors expect other problems if the present method is applied for a structure similar to this from ref. 41? I think it would be fair to the readers to discuss these and other potential problems with thermal control of BZ LMs.

Authors: We have added the following in the Discussion section:

To control frequencies in an ensemble of BZ LMs~\cite{fullarton2018belousov} we can use small Peltier elements, which size is enough just to fit a single marble, e.g. Peltier Module, 1.3~W, 2.2~A, 0.9~V, 6\$times\$6~mm (each of the modules can be automatically controlled via Arduino device).

3. As described in caption of Fig. 2c in this experiment oscillations were induced with a silver wire. Please add information about purpose of silver wire stimulation to Methods section.

Authors: we added

(reaction starts because a concentration of bromate is reduced locally due to ionisation of silver)

4. In Fig. 3d it would be helpful if the moments of time at which Peltier was on and off are also marked (as in Fig 3a with * and # symbols).

Authors: Done.

5. There are some small issues, which I've noticed, and which could help the Authors with polishing the manuscript:

- phrase repetitions

line 36 page 5 “are as follows are analysed”.

Line 43 page 7 “This is is reflected”

- According to caption of Fig.4a it was kept cool until the end of experiment. Why # symbol is visible at t = 900 s?

Authors: corrected.